# Supramarginal Gyrus and Angular Gyrus Subcortical Connections: A Microanatomical and Tractographic Study for Neurosurgeons

**DOI:** 10.3390/brainsci13030430

**Published:** 2023-03-02

**Authors:** Mehmet Hakan Şahin, Mehmet Emin Akyüz, Mehmet Kürşat Karadağ, Ahmet Yalçın

**Affiliations:** Neurosurgery Depertmant, School of Medicine, Ataturk University, 25100 Erzurum, Turkey

**Keywords:** angular gyrus, fiber dissection, microsurgical anatomy, supramarginal gyrus, white matter anatomy

## Abstract

**Background and Objectives:** This article aims to investigate the subcortical microanatomy of the supramarginal gyrus (SMG) and angular gyrus (AnG) using a microfiber dissection technique and diffusion tensor imaging (DTI)/fiber tractography (FT). The cortical and subcortical structures of this region are highly functional, and their lesions often present clinically. For this reason, the possibility of post-surgical deficits is high. We focused on the supramarginal gyrus and the angular gyrus and reviewed their anatomy from a topographic, functional and surgical point of view, and aimed to raise awareness especially for neurosurgeons. **Methods:** Four previously frozen, formalin-fixed human brains were examined under the operating microscope using the fiber dissection technique. Four hemispheres were dissected from medial to lateral under the surgical microscope. Brain magnetic resonance imaging (MRI) of 20 healthy adults was examined. Pre-central and post-central gyrus were preserved to achieve topographic dominance in dissections of brain specimens. Each stage was photographed. Tractographic brain magnetic resonance imaging of 10 healthy adults was examined radiologically. Focusing on the supramarginal and angular gyrus, the white matter fibers passing under this region and their intersection areas were examined. These two methods were compared anatomically from the lateral view and radiologically from the sagittal view. **Results:** SMG and AnG were determined in brain specimens. The pre-central and post-central gyrus were topographically preserved. The superior and medial temporal gyrus, and inferior and superior parietal areas were decorticated from lateral to medial. U fibers, superior longitudinal fasciculus II (SLF II), superior longitudinal fasciculus III (SLF III), arcuat fasciculus (AF) and middle longitudinal fasciculus (MdLF) fiber groups were shown and subcortical fiber structures belonging to these regions were visualized by the DTI/FT method. The subcortical fiber groups under the SMG and the AnG were observed anatomically and radiologically to have a dense and complex structure. **Conclusions:** Due to the importance of the subcortical connections of SMG and AnG on speech function, tumoral lesions and surgeries of this region are of particular importance. The anatomical architecture of the complex subcortical structure, which is located on the projection of the SMG and AnG areas, was shown with a DTI/FT examination under a topographic dominance, preserving the pre-central and post-central gyrus. In this study, the importance of the anatomical localization, connections and functions of the supramarginal and angular gyrus was examined. More anatomical and radiological studies are needed to better understand this region and its connections.

## 1. Introduction

Neuroradiology and neuroanatomical studies have been gaining importance in recent years in neurology, neurosurgery and basic neuroscience in understanding white matter structure [1].

Cerebral subcortical anatomy and white matter pathways are a matter of great interest to scientists. Related to this, white matter anatomy studies were conducted in the first half of the 20th century [2]. Today, white matter dissections in brain specimens are performed with precision dissections, after a 1-month fixation with 5% formalin, which was defined by Klinger, and then kept at −10 degrees for 1 week [2]. Learning the white matter pathways that connect cortical areas to each other is crucial to understanding the anatomical components of health or disease. Diffusion tensor imaging (DTI) shows these pathways non-invasively, but due to intersecting fibers, more specific solutions are needed. The solution to this problem is to show specific paths with single magnetic resonance imaging (MRI) voxels. The recent advent of DTI/FT imaging has allowed the investigation of these pathways in vivo [3].

The presence of diseases affecting a number of white matter structures, such as multiple sclerosis, showed that cerebral white matter has a wide clinical significance. Diffusion tensor imaging/fiber tractography (DTI/FT) studies have become increasingly important to understand cortical–subcortical connections and their correlation–integration high brain functions. Due to the limitations of DTI/FT studies, microanatomical dissections are still the gold standard method for confirming radiological data [4]. The combination of microanatomical dissections and DTI/FT studies is necessary for a better understanding of these complex subcortical structures in 3D.

Understanding this 3D architecture has a special importance for neurosurgeons. A good knowledge of neuroanatomy and neuroradiology is required to plan the most optimal surgery, especially in brain tumors where the normal anatomical structure changes frequently. When brain tumors expand from the point of origin, they do not pass through the subcortical connections, and push the white matter fibers out of their localization. This is also demonstrated in DTI/FT studies. For this reason, it is important for neurosurgeons to dominate these and similar areas with DTI/FT studies and microanatomical dissection studies when planning surgery [5].

General concepts of language were devised by Broca and Wernicke. Recent models show that more complex connections and more cortical/subcortical structures participate in this function rather than general concepts [6].

The general knowledge in these speech paths is the opinion that the left side is dominant, but studies have shown that both sides can play an active role in listening to speech. However, more specific studies are needed on this function of both hemispheres [6,7,8].

The supramarginal gyrus (SMG) and the posterior superior part of the temporal lobe, which we want to focus on, are important regions for the language of the white matter pathways [1]. Our aim was to focus on the SMG and angular gyrus AnG, which are important areas for speech paths, and to examine the subcortical connections (superior longitudinal fasciculus II (SLF II), superior longitudinal fasciculus III (SLF III), arcuat fasciculus (AF), middle longitudinal fasciculus (MdLF)) of this region only microanatomically and through the DTI/FT by preserving the pre-central and post-central gyrus, creating a topographic dominance, and to provide a perspective to neurosurgeons.

For this purpose, four brain specimens obtained from the American Tissue Bank were subjected to microanatomical dissections with the Klinger method, and the images obtained from the MR tractography images of 10 healthy individuals were examined.

The pre-central gyrus and post-central gyrus were preserved and white matter pathways important for language were shown without losing topographic dominance. Parallel matching of dissections was achieved by showing these pathways in healthy individuals tractographically with known techniques. Anatomical, functional and surgical information was compiled for a better understanding of this region.

## 2. Materials and Methods

This study was approved by our local ethics committee (B.30.2.ATA.0.01.00/659) and carried out according to the 1964 Helsinki declaration. Due to the retrospective nature of the study, no explicit informed consent was obtained from subjects.

Brains were obtained from the American Tissue Bank after ethics committee approval. The causes of death are reported in their tags, as donors do not have an intracranial pathology. Four formalin-fixed human brains were prepared in accordance with the Klinger method, after being kept in 5% formalin for 1 month and then frozen at −20 °C for 1 week^2^. The arachnoid and vascular structures of these four brains were removed. Then, these specimens were kept in the freezer at −20 degrees for one week. Frozen brains were kept in room temperature water and thawed. A high-magnification operating microscope (ZEISS OPMI-MD S3, Oberkochen, Germany), wooden spatula, watchmaker forceps and other surgical instruments were used throughout the dissection. Topographically, classical landmarks in the cerebral hemisphere were determined and lateral to medial dissections were made gradually along the anatomical axes. By preserving the pre-central and post-central gyrus, 3D anatomical dominance was achieved, and the white matter fibers passing under the SMG/AnG were focused and dissection was performed. For the radiological evaluation, 10 healthy individuals were selected in groups. Healthy individuals were older than 18 years of age and there was no gender discrimination between male and female. A specific selection was not made, as an age- and sex-related variation in the examined white matter pathways was not reported in the literature. The sagittal sections of brain magnetic resonance images (MRI) of 10 people were analyzed to show cortical anatomical structures. SLF II, SLF III, AF and MdLF white matter pathways passing in the subcortical level of the AnG and SMG were shown in these 10 individuals who underwent DTI/FT examination. In this study, it was aimed to develop a more understandable anatomical perspective by lateral dissection of brain specimens and examining tractographic images from the sagittal view.

All brain DTI/FT examinations were performed using a 3T MR scanner (Skyra, Siemens Healthcare, Erlangen, Germany). A 20-channel dedicated head coil was used in all examinations. An anatomical image was constructed using sagittal T1 3D MPRAGE with the following parameters: TR, 1420 ms; TE, 3.9 ms; FOV, 25 × 25 cm; voxel size, 1 × 1 × 1 mm. Tensor tractography imaging consisted of axial epiplanar imaging with the following parameters: TR, 3900 ms; TE, 95 ms; EPI factor, 30; voxel size, 2 × 2 × 4 mm; and a number of diffusion directions, 64. Diffusion trace images at b = 1000 s/mm^2^ along with b0 images were acquired and tensor maps covering the whole brain were calculated automatically. Fiber tracking and tensor data were analyzed using an offline workstation (Syngo via, Siemens Healthcare, Erlangen, Germany). Anatomical images consisting of 3D volumetric T1 weighted images were fused with tensor tractography before evaluation. Probabilistic fiber tracking was used to determine the main white matter fiber tracts. AF, SLF II, SLF III and MdLF tracts were reconstructed using VOI-based deterministic fiber tracking on previous fiber tracts depicted via probabilistic fiber tracking. Curved multiplanar reconstruction and 3D modeling were used to delineate the fiber tracts.

## 3. Results

### 3.1. Fiber Dissection Procedure: Lateral to Medial Dissection

The right hemisphere lateral surface sulcus and gyrus structures were easily seen. SMG, AnG, pre-central gyrus, post-central gyrus, triangular gyrus, opercular gyrus, inferior and middle frontal gyrus, superior and middle temporal gyrus and occipital lobe were seen (Figure 1A). The pre-central gyrus, post-central gyrus and triangular gyrus were topographically preserved in order not to lose the sense of depth during dissection. Afterwards, the inferior frontal gyrus, SMG, AnG and superior temporal gyrus were decorated, and U fibers were seen (Figure 1B). The walls and U fibers of these gyrus were removed and the closely related SLF II, SLF III and AF white matter fiber groups were outlined (Figure 1C). Afterwards, the inferior frontal gyrus, SMG, AnG and superior temporal gyrus were decorated, and U fibers were seen. The walls and U fibers of these gyrus were removed and the closely related SLF II, SLF III and AF white matter fiber groups were outlined. After precise microanatomical dissection, the AF and SLF II white matter fiber groups and SLF III white matter fiber groups located immediately dorsolaterally in this group were seen. In order to better understand the fiber orientations in these groups, the middle temporal gyrus and superior parieal area were decorticated, and the U connections between the C-shaped fibers of SLF II, SLF III and AF extending from temporoparietal to frontal and the surrounding gyrus were more clearly revealed. The U fibers in the surrounding tissues were removed. The borders of SLF II, SLF III and AF in the inferior parietal region were clearly defined (Figure 1D). The C-shaped or horseshoe-shaped orienting fibers of this region were separated into layers by microdissection. Finally, dissection was continued at the level of the posterior insular point. The fibers of MdLF, which receives fibers from the posterior part of the superior and medial temporal gyrus, were shown in close association with the inferior–medial border of the AF. Thus, the closely related connections of SLF II, SLF III, AF and MdLF at the level of the AnG and the SMG were demonstrated, along with the topographic preservation of the pre-central gyrus, post-central gyrus and triangular gyrus for depth perception (Figure 1E).

### 3.2. Radiological Examination

In all examinations, the complex pathway of AF and SLF II and III showed continuity between the posterior superior/middle temporal gyrus and the frontal lobe. While SLF II fibers moved from the ventrolateral to middle frontal gyrus, SLF III fibers moved from the dorsolateral to frontal operculum. MdLF fibers intertwined with the AF and SLF complex (SLF II-III) and showed a vertical course to this group and showed continuity from the superior temporal gyrus to the superior parietal lobe. The densest intersection of AF, SLF II, SLF III and MdLF fibers corresponded to the SMG and AnG projection (Figure 2A,B).

## 4. Discussion

### 4.1. Topographic Anatomy

When the brain specimen is viewed from the lateral side, the lateral surfaces of the frontal, parietal and occipital lobes are seen. Variations in the structure of the sulcus and gyrus can be seen from person to person and even in different hemispheres in the same person. The sylvian fissure and central sulcus are seen in nearly 100% of individuals. These two sulci are the most important landmark areas on the lateral surface of the brain. The superficial part of the sylvian fissure is easily visible from the lateral part of the brain. The superficial part consists of a stem and three rami. The posterior ramus is the longest part. Superiorly, it separates the frontal and parietal lobes from the temporal lobe. Its most posterior border ends in the SMG.

The operculoinsular region consists of two narrow narrows, the operculum and the insula. The opercular narrow upper limbi form the upper edge of the sylvian fissure. From anterior to posterior, they form the pars orbitalis, triangularis, opercularis, pre-central, post-central and SMG, respectively [9].

The AnG is located posterior to the superior temporal sulcus. The AnG and the SMG in front of the angular gyrus together form the inferio-parietal lobe. These two gyrus are interconnected by horizontal U fibers [9,10]. AnG is a structure belonging to the parietal lobe. Due to the sulci on this structure, which forms the posterior border of the superior temporal gyrus, it can be seen as 2–3 pieces [9]. At the posterior end of the sylvian fissure, there are two arches: the SMG anteriorly and the AnG posteriorly. The SMG is adjacent to the post-central sulcus anteriorly and to the superior temporal gyrus inferiorly, and it is in continuity with these two structures [11].

There are studies showing that it is supported by the superior temporal sulcus, insula cortex and amygdala/hippocampus in functional MRI studies for understanding emotion recognition [12,13]. The AnG receives white matter connections from the classic language regions: the inferior frontal and superior temporal cortex. However, this region has complex and large dentric connection areas [11].

Subcortical fiber groups that provide connections with other cortical centers pass under the AnG and SMG. The closest subcortical fibers to this region are the SLF II, SLF III, AF and MdLF [14].

The morphology of superior longitudinal fasciculus (SLF) and AF has been investigated since the 19th century [2]. There are many publications that use the same nomenclature for SLF and AF [15,16,17]. However, Schmahmann et al.’s study in rhesus monkeys divided SLF into four parts. In this study, these four segments were named as three horizontal segments numbered superior longitudinal fasciculus I(SLF-I), SLF-II, SLF-III and AF [18]. In studies with alternative nomenclature, it has been accepted that these fiber groups are separate anatomical structures [1].

These nomenclature studies have divided SLF (such as SLF-I, SLF-II, SLF-III) and AF (such as superior longitudinal fasciculus IV or vertical AFv, horizontal AFd) into different sections [19,20,21,22]. In our study, we used SLF-I, SLF-II, SLF-III and AF nomenclature used by Markis et al. [20].

SLF-I runs from the region between the medial parieto-occipital and medial frontal over the cingulate sulcus. SLF-I is adjacent and parallel to the cingulum. It occurs in the dorsal–medial region of both hemispheres [20,23].

The horizontal portion of SLF-II and AF is rostral to the lateral sulcus and lateral to the corona radiata. SLF-II and AF contain groups of white matter fibers between the middle frontal gyrus, the dorsal part of the pre-central gyrus and the AnG [20,23].

SLF-III starts from the supramarginal gyrus. SLF-III fibers pass through the parietal and frontal opercular white matter with a ventral and lateral orientation. SLF-III reaches the dorsal part of the pre-central gyrus and the inferior frontal gyrus (pars opercularis) [20,23].

Makris et al. divided AF into two parts based on orientation: vertical (AFv) and horizontal (AFd). The vertical (AFv) extends from the caudal part of the superior temporal gyrus to the lateral of the posterior horn of the lateral ventricle. It extends from the horizontal (AFd) temporo-occipital transition region to the lateral pre-frontal cortex adjacent to the SLF-II [20].

MdLF was first described in macaque monkeys by Seltzer and Pandya [24]. The existence of this pathway in humans was first reported by Makris et al. and demonstrated by tractography studies [25]. MdLF was thought to be a subcortical white matter pathway connecting the superior temporal gyrus to the AnG, but as a result of tractographic and anatomical studies, MdLF was found to have a more complex structure [4]. In tractography and microanatomical studies, there are differences in terms of morphology, orientation and nomenclature [26,27,28,29]. MdLF is located medial to the SLF/AF complex under the parietal and occipital lobes. MdLF fibers course very closely posterior to the superior temporal gyrus, then medial to the “C”-shaped fiber group of the AF. Therefore, the microanatomical dissection of this complex area is also difficult [4].

Kalyvas et al. divided MdLF into three parts. MdLF-1 is the white matter fibers between the dorsolateral superior temporal gyrus and the precuneus, and MdLF-2 is between the dorsolateral superior temporal gyrus and the parieto-occipital area. MdLF-3 is the white matter fibers between the superior temporal gyrus and the occipital area from the lower level of AG [4].

### 4.2. Functional Anatomy

Speech and language are unique features that distinguish humans from other living things. Animals have communication systems among themselves, but not comparable to humans. From this point of view, physiological and anatomical studies have been carried out investigating the communication system in humans. The interrelationships of different cortical regions in the perception and processing of speech are still under investigation. As far as is known, the frontal, temporal and parietal regions and their connections are the dominant areas in this feature [7]. In addition to the classical language organizations of the Broca and Wernicke areas, the superior temporal gyrus and inferior parietal lobe (SMG and AnG) and white matter connections are of great importance [7].

Gerstmann syndrome findings (acalculia, finger agnosia, right–left disorientation and agraphia) are seen in a subcortical isolated lesion, especially in the AnG and SMG areas in the dominant hemisphere [30]. Cortical and subcortical regions are interconnected and are known to transmit information between different regions. The AnG and the SMG (especially the dominant side) are in a region that governs important functions such as language, arithmetic operations, written expression and right and left discrimination [30].

AnG is thought to be a cognitive center that provides conceptual meaning thanks to its connections with different centers such as posterior middle temporal gyrus, anterior temporal region, inferior frontal region and pre-supplementary motor area regions [31]. It is also thought that the right and left SMG have different functions. For example, the left SMG is thought to be responsible for tool use and language skills. It is thought that the right SMG may have an effect on emotion recognition ability [31].

The AF and SLF are the main groups of fibers involved in the dorsal tract transmission of language processing. Among the subgroups of SLF, SLF II and III are important in language processing. Little is known about the functional role of MdLF. Some authors consider a connection or potential pathway between the dorsal and ventral passages in the language network. It is suggested that MdLF is related to language comprehension, visuospatial and attention functions and integration between auditory and visual information [4,27,29].

It is known that the classical language organization consists of the inferior frontal gyrus (Broca’s area), the superior temporal gyrus and the AF, which provides the connections between this region. Along with functional and neuroimaging methods such as DTI/FT, a more detailed idea has begun to emerge about the functional neuroanatomy of language with the developments in stimulation mapping techniques. Cortical and subcortical connections in phonological processes are important for modern network-based speech models. For this reason, the language network contains a more complex network rather than basic knowledge. Despite recent developments, many questions still await answers [6].

It is thought that there are two paths for the dorsal stream path, and these paths are called dual streams. These pathways include the posterior frontal lobe and sylvian parieto-temporal region connection, and the dorsal superior temporal gyrus and temporo-parietal region connection. The common crossing point of these two roads coincides topographically with the lower level of the AnG and the SMG [6].

### 4.3. Surgical Perspective

This is important in planning the surgery of this particular area. In studies in which two patient series were compared, it was shown that the post-operative neurological status of the operations performed with subcortical mapping and pre-operative DTI/FT had better results [32,33].

The correlation between pre-operative DTI/FT and intraoperative subcortical stimulation is variable and appears to be highly sensitive. The sensitivity varies according to the white matter fiber group, and high sensitivity is found for the SLF pathway. An important point here is to determine the resection margins and resection technique, considering that the white matter tracts may shift during surgery. As a surgical technique, it is recommended to establish a safety margin of 5 mm and to start resection near the subcortical pathways [34,35,36]. Tumor type is also important in surgery. While high-grade gliomas tend to push or displace the white matter pathways, low-grade tumors can be seen to grow embedded in the white matter pathways [32,33].

In a study, Chang et al. made recommendations for the surgeries of auditory pathways. Chang et al. said that functional MRI and DTI/FT give an idea for the determination of subcortical anatomy and localization of dominant functionals, but they should not be used in isolation as they may be insufficient in terms of details. They also report that cortical and subcortical mapping should be used when working in subcortical regions known to be involved in language processing. It should be noted that all these techniques significantly reduce post-operative neurological deficits; however, patients may develop a probable neurological deficit at a high rate [6].

Studies on language in recent years have revealed that it has a complex structure and that much research is necessary. With an innate ability, the brain has the ability to carry its current information from the areas where it has lesions. There are studies supporting this mechanism, which is explained with the “hodotropy” paradigm [37,38,39]. Along with knowing this plasticity feature, knowledge of cortical and subcortical anatomy constitutes the most important weapon of neurosurgeons [6].

## 5. Conclusions

This is the first study to provide a better understanding of the anatomical architecture of the supramarginal gyrus and angular gyrus topographically, while preserving the post-central and pre-central gyrus adjacent to the inferior parietal region. The SMG, AnG and subcortical connections (SLF, AF, MdLF) have important functions. The most important weapon of the neurosurgeon for the most optimal surgery of this region is to have command of the topography of this region, microanatomical dissections and good radiological (such as DTI/FT) imaging. More microanatomical dissection and radiological studies are needed to understand the connections of this region.

## Figures and Tables

**Figure 1 brainsci-13-00430-f001:**
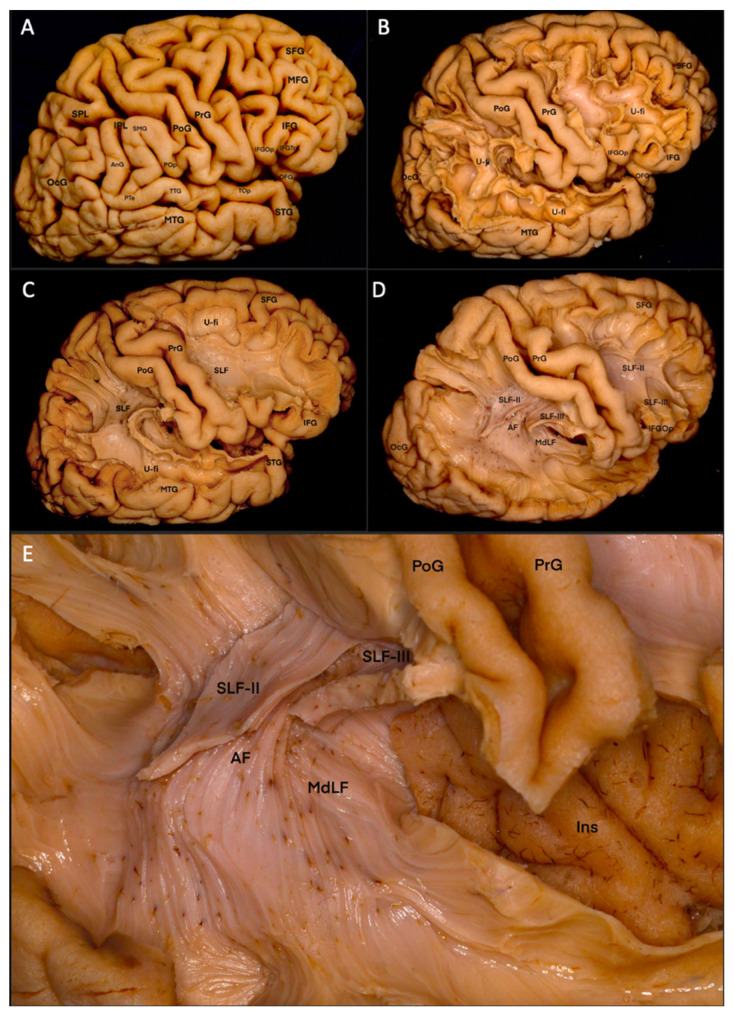
(**A**): When we look at the lateral surface of the hemisphere of the brain, AnG, SMG, PrG, PoG and surrounding anatomical structures appear. (**B**): View of U fibers after decorticating the inferior frontal gyrus, supramarginal gyrus, angular gyrus and superior temporal gyrus. (**C**): The walls and U fibers of these gyrus were removed and the closely related SLF (SLF II- III) was seen. (**D**): After the precise microanatomical dissection, AF and SLF II white matter fiber groups and SLF III white matter fiber groups located immediately dorsolaterally in this group were seen. (**E**): The focus was on the area with the ANG and SMG projection. Microanatomical dissection was deepened. The intersection of MdLF with the AF and SLF complex was seen. AF, arcuat fasciculus; AnG, angular gyrus; IFG, inferior frontal gyrus; IFGOp, inferior frontal gyrus opercular part; IFGTr, inferior frontal gyrus triangular part; Ins, insula; IPL, inferior parietal lobule; MdLF, middle longitudinal fasciculus; MFG, medial frontal gyrus; MTG, medial temporal gyrus; OcG, occipital gyri; OFG, orbitofrontal gyri; PoG, post-central gyrus; PrG, pre-central gyrus; Pte, planum temporale; POp, parietal operculum; SFG, superior frontal gyrus; SLF, superior longitudinal fasciculus; SLF II, superior longitudinal fasciculus II; SLF III, superior longitudinal fasciculus III; SMG, supramarginal gyrus; SPL, superior parietal lobule; STG, superior temporal gyrus; TOp, temporal operculum; TTG, transverse temporal gyri; U-fi, U fibers.

**Figure 2 brainsci-13-00430-f002:**
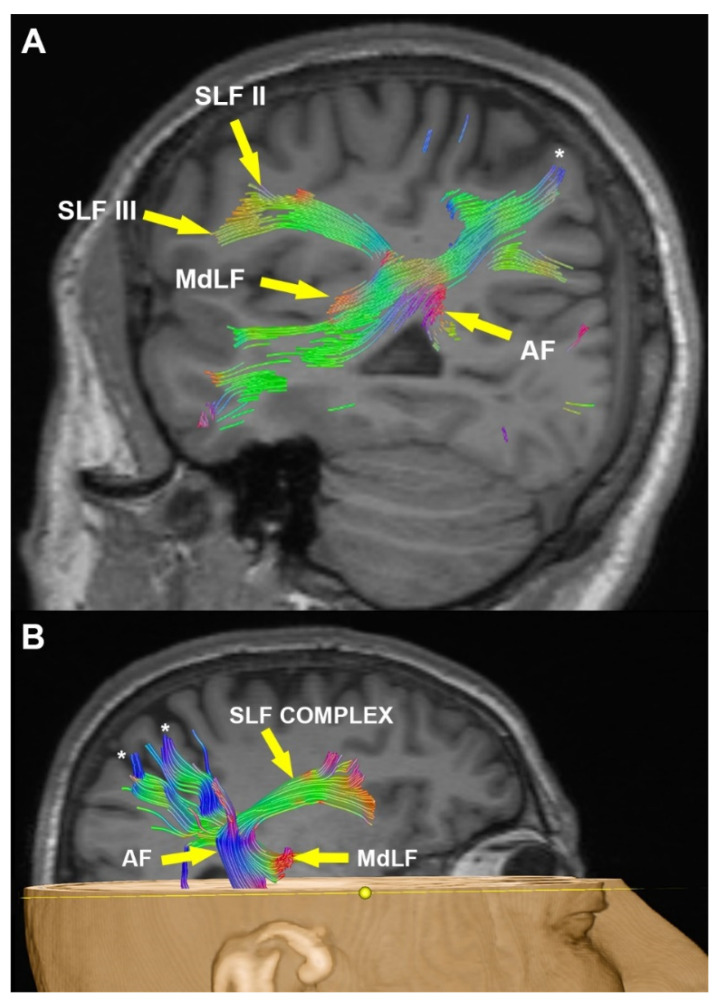
Sagittal (**A**) and 3D volume rendered (**B**) view of T1 weighted MR image with fused tractography show the complex course of AF, SLF II, SLF III and MdLF together (arrows). Note the superior oblique course of MdLF to the precuneus (asterisks). SLF II and III fibers together marked as “SLF COMPLEX”.

## Data Availability

The data used in the article will be shared by the responsible author.

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
