# Peer review of "Supramarginal Gyrus and Angular Gyrus Subcortical Connections: A Microanatomical and Tractographic Study for Neurosurgeons"

_brainsci, 2023, doi:10.3390/brainsci13030430_

Round 1

Reviewer 1 Report

The authors investigated the subcortical microanatomy of the supramarginal gyrus (SMG) and angular gyrus (AnG) using microfiber dissection technique and diffusion tensor imaging (DTI)/fiber tractography (FT).

Abstract: the introduction part (background) of the abstract is not well introduced. Why these brain parts are important should be explained more in detail here. Not in the conclusion part.

Introduction part: very confusing, there is no proper flow. The reader cannot properly see why the authors have chosen these brain areas. DTI/FT are also not properly explained.

All abbreviations should be explained with whole words again, not only in the abstract.

Author Response

Comment 1: Abstract: the introduction part (background) of the abstract is not well introduced. Why these brain parts are important should be explained more in detail here. Not in the conclusion part.

Answer 1: The abstract part has been rearranged.

Comment 2: Introduction part: very confusing, there is no proper flow. The reader cannot properly see why the authors have chosen these brain areas. DTI/FT are also not properly explained.

Answer 2: Introduction part has been reorganized.

Comment 3: All abbreviations should be explained with whole words again, not only in the abstract.

Answer 3: Abbreviations have been rearranged.

Reviewer 2 Report

Minor:

Intro

- Please define acronyms for the reader in the intro section not at the end, example in this sentence:

Our aim was to focus on the SMG and AnG, which is an important area for speech 56 paths, and to examine the subcortical connections (SLF II, SLF III, AF, MdLF) of this region 57 only microanatomically and DTI/FT

- Please clearly state the hypothesis of the study.

- Overall the intro is really short and doesn't really help to understand the background and the purpose of the study. I would suggest to develop more the introduction section to provide more information about the background and the purpose of the study.

Materials methods

"Four formalin-fixed human brains" How did you select those brains?

"The arachnoid and vascular structures of these six brains were removed" Where those 6 brains come from?

"Dissections from lateral to medial were performed gradually along the 69 anatomical axes." what did you dissect exactly? a specific region? 

Maybe 3.1 should be included in materials and methods

Overall, materials and methods section is misleading and not enough detailed. It's not clear why you compare dissected brain with a cohort of alive patients. What demographical population did you choose for the study ? man, female. You state that the patients are healthy, what about the brains you used, are they from unhealthy people? are they age-matched? 

What are your inclusion/exclusion criteria?

Results 3.2: What is the method to draw the fibers? What is the point of this result? 

This MS is too misleading and not really clear. Introduction, materials and methods as well as results sections should be re-written for more clarity. Also, I don't understand the purpose of this study. Authors should clarify that. 

Author Response

Comment 1:  Please define acronyms for the reader in the intro section not at the end, example in this sentence

Answer 1: Abbreviations have been rearranged.

Comment 2: Please clearly state the hypothesis of the study

Answer 2: The abstract part of the study was rearranged and our hypothesis was clarified.

Comment 3:  Overall the intro is really short and doesn't really help to understand the background and the purpose of the study. I would suggest to develop more the introduction section to provide more information about the background and the purpose of the study.

Answer 3: Introduction part has been reorganized.

Comment 4: Four formalin-fixed human brains" How did you select those brains?

The arachnoid and vascular structures of these six brains were removed" Where those 6 brains come from?

Answer 4: The four brains we used in our study were obtained from American tissue banks through intermediary institutions in our country. Each specimen has tags containing its own information and it is stated that there is no intracranial pathology among the causes of death of the individuals. We added it to the material and method section on this comment. If the editor requests, the tags of the brain specimens will be shared on condition that they remain confidential.

Comment 5: "Dissections from lateral to medial were performed gradually along the 69 anatomical axes." what did you dissect exactly? a specific region?

Answer 5: Lateral to medial dissection is the classic term for white matter anatomical dissections. Our focus is the inferior parietal lobule, which is the junction of SMG and AnG, and the underlying white matter fibers. For this reason, the inferior frontal lobe, inferior parietal lobe and superior temporal gyrus, where the white matter fibers extend, were subjected to decortication. Precentral and postcentral gyrus are preserved for topographic dominance.

Comment 6: Overall, materials and methods section is misleading and not enough detailed. It's not clear why you compare dissected brain with a cohort of alive patients. What demographical population did you choose for the study ? man, female. You state that the patients are healthy, what about the brains you used, are they from unhealthy people? are they age-matched?

Answer 6: Matching white matter pathways with tractographic methods is a popular method in current literature publications. The aim here is to facilitate the radiological matching of the anatomically shown white matter pathways in the reader's memory. In particular, it is aimed that surgeons preparing for surgery better understand radiology. The gender and age-related variational structure of the pathways we demonstrated in white matter studies in adult patients is not found in the literature. For this reason, we did not discriminate between the adult individuals we selected by age and gender. It is reported in tissue bank documents that the cause of death of the brain specimens we used was not due to intracranial pathology. It is a detail taken into consideration by the relevant institution when brain specimens for neurosurgery are obtained from tissue banks.

The material and method section has been edited on this comment.

Comment 7: What is the method to draw the fibers? What is the point of this result? 

Answer 7:  This method of white matter studies is a method of selecting ROI points and tracking focal fibers with it. These are the drawings produced by radiology clinics with special techniques accompanied by optional programs of MR devices and are frequently seen in the literature. Explanations for DTI/FT have been expanded upon this comment.

Comment 8: This MS is too misleading and not really clear. Introduction, materials and methods as well as results sections should be re-written for more clarity. Also, I don't understand the purpose of this study. Authors should clarify that.

Answer 8: On this comment, adjustments were made to the introduction, material and method parts. The aim of the study was explained more clearly by making additions in the abstract and introduction part.

Round 2

Reviewer 1 Report

The authors have improved their paper.

Author Response

thank you for your contributions

Reviewer 2 Report

Thanks for the changes

Author Response

thank you for your contributions